# Effects of Simulated Heat Waves on Life History Traits of a Host Feeding Parasitoid

**DOI:** 10.3390/insects10120419

**Published:** 2019-11-22

**Authors:** Yi-Bo Zhang, An-Pei Yang, Gui-Fen Zhang, Wan-Xue Liu, Fang-Hao Wan

**Affiliations:** 1State Key Laboratory for Biology of Plant Diseases and Insect Pests, Institute of Plant Protection, Chinese Academy of Agricultural Sciences, Beijing 100193, China; zhangyibo@caas.cn (Y.-B.Z.); yap2002@126.com (A.-P.Y.); zhangguifen@caas.cn (G.-F.Z.); liuwanxue@caas.cn (W.-X.L.); 2Scientific Observing and Experimental Station of Crop Pests in Guilin, Ministry of Agriculture, Guilin 541302, China; 3Institute of Plant Protection, Xinjiang Academy of Agricultural Science, Urumqi 830091, China

**Keywords:** heat wave, amplitude, frequency, life history traits, parasitoid

## Abstract

The frequency and amplitude of heat waves are predicted to increase under future climate change conditions. We still lack a detailed understanding of how changes in the frequency and amplitude of heat waves are linked to the life history traits and biocontrol efficiency of host-feeding parasitoids. In the present study, we simulated a series of heat waves as a function of amplitude and frequency to investigate the effects on the life history traits of the host-feeding parasitoid *Eretmocerus hayati*. We found that both the amplitude and frequency of heat waves significantly affected the adult phenotypes. In the low-amplitude heat wave group, the frequency of heat waves did not change the life history traits of the parasitoid; however, when the heat amplitude reached 42 °C, medium (four times/week) and high frequencies (seven times/week) of heat waves detrimentally affected these parameters. Hence, these findings suggest that to obtain optimal biological control with this parasitoid, we need to carefully monitor heat wave pattern (especially the amplitude and frequency) over the short term (usually 7–10 days) before releasing a host-feeding parasitoid.

## 1. Introduction

Climatic changes, such as variable mean temperatures and extreme temperature events, play a pivotal role in the distribution ranges and communities of species [1,2]. Although both mean temperatures and extreme temperatures can have significant effects on organisms and ecological interactions, most studies have biasedly focused on the effects of variable mean temperatures on species fitness or life history traits, thereby neglecting extreme temperature events [2,3]. In fact, extreme temperature events, especially heat waves, will become increasingly frequent and will have increased amplitudes [4]. Moreover, their impacts on organisms are thought to be more important than those associated with mean temperatures [2].

Extreme temperature events (heat waves) have the potential to affect insect phenology and population parameters because insects, as typical poikilothermic animals, do not have physiological mechanisms for regulating their internal temperature [2,3]. However, heat waves have different effects on life history traits and the fitness of insects depending on the heat amplitude and heat frequency. For example, heat waves within the optimal thermal zone showed a positive relationship between the amplitude of the heat wave and egg production in *Ceratitis capitate* (Wiedemann) [5], whereas heat waves involving stressful temperatures reduce the fecundity and longevity of the tortricid moth *Zeiraphera canadensis* Mutuura and Freeman [6], *Drosophila melanogaster* Meigen [7] and *Diadegma semiclausum* Hellén [8]. To date, many studies have focused on how heat waves affect the phenotype of primary herbivorous insects [5,6,7], and few studies have focused on secondary parasitic insects [8,9,10]. With increasing chemical insecticide resistance in pests, biological control strategies using parasitic natural enemies as an alternative measure have been extensively applied in agriculture. Parasitoids are insects (usually wasps or flies) that lay eggs inside or on other insects (the host) [11,12]. To successfully parasitize a host, female parasitoids must find a host patch, locate the host inside the patch and identify hosts that will be profitable for the development of the offspring [12]. Hence, understanding how a parasitoid copes with extreme heat waves and whether there is a negative effect on its life history traits and biocontrol efficiency are of great importance. *Eretmocerus hayati* Zolnerowich and Rose (Hymenoptera: Aphelinidae), a host-feeding primary parasitoid of *Bemisia tabaci*, is widely used as a biological control agent to control whiteflies in China, as it not only parasitizes the nymphs of whiteflies but also feeds directly on its host [13,14]. To improve the mass-rearing efficiency of this biocontrol species, we confirmed that switching temperatures during the immature and adult stages can greatly improve life table parameters, especially the intrinsic rate of increase [15]. This study significantly increased the efficiency of mass rearing *E. hayati* indoors. However, a desired biological control agent should not only be mass reared in the laboratory but also be able to endure variable outdoor environments [3,9,10]. No study has considered how heat waves with different amplitudes and frequencies affect *E. hayati* and whether there is a negative effect on its life history traits and adult biological control efficacy.

In this study, we investigated the effects of experimental heat wave amplitude and frequency on the life history traits (host-feeding events, fecundity longevity and total host mortality) and biocontrol efficiency of a host-feeding parasitoid, *E. hayati*. Our goals were to investigate the effects of heat waves with varying amplitudes and frequencies on the life history traits of a host-feeding parasitoid to evaluate its biocontrol efficiency and offspring fitness.

## 2. Material and Methods

### 2.1. Whitefly and Parasitoid Cultures

All plants and insects were maintained in a climate-controlled chamber at 26 ± 1 °C with 70–80% relative humidity (RH) and a 14 h:10 h light: dark (L:D) photoperiod at the Department of Biological Invasions (DBI), Institute of Plant Protection, Chinese Academy of Agricultural Sciences, Beijing, China.

Colonies of *B. tabaci* cryptic species Mediterranean (MED, formerly Q biotype) were isolated and maintained on young cotton plants (*Gossypium hirsutum* L.) in a plastic box (20 × 30 × 20 cm). The cotton plants were planted in black turfy soil and were used to culture the whitefly colony when four true leaves had fully developed. The species identity of the whiteflies in the colony was checked each month by sequencing the COI mtDNA from the whiteflies and then comparing it with the sequences available in the NCBI database.

To generate a laboratory colony, *E. hayati* was collected from cotton fields during the summer of 2012 in Ha-Mi region of Xinjiang Uygur Autonomous Region, northwestern China (E89°8′, N42°53′, 12 m a.s.l.). The colony was then reared on 2nd–3rd-instar nymphs of *B. tabaci* collected from the same location as the parasitoids maintained on young cotton plants in the greenhouse of the DBI. Thereafter, the species was identified by phylogenetic analysis based on COI sequences and cross-testing between this species and known *E. hayati* [16]. Moreover, to avoid inbreeding, new parasitoid individuals were captured from the field every year and were released in our indoor colonies.

### 2.2. Experimental Setup

A two-factorial experiment was established to explore the effect of amplitude and frequency of extreme fluctuating temperature regimes (i.e., heat waves) on the life history traits of *E. hayati*. Based on 5 years of field observations, we found that 42 °C was the most frequent extreme temperature in Ha-Mi region of Xinjiang Uygur Autonomous Region. Hence, we chose 42 °C and 36 °C as the high and low amplitude temperatures of the heat waves. The frequency of heat waves was divided into three levels: high frequency (seven times/week), medium frequency (four times/week), and low frequency (one time/week). The temperature regimes generated by the combination of amplitude and frequency (Figure 1) were (1) High amplitude–High frequency (HaHf), with one peak at 42 °C per day. On the days with a peak, the temperature started to increase at 12:00, reached a maximum from 12:00 to 12:30 that was maintained for 2 h, and then decreased at 14:30. The temperature was kept constant at 26 °C for the rest of the day. The other regimes were (2) High amplitude–Medium frequency (HaMf), with four peaks of 42 °C per week; (3) High amplitude–Low frequency (HaLf), with one peak at 42 °C per week; (4) Low amplitude–High frequency (LaHf), with one peak at 36 °C per day; (5) Low amplitude–Medium frequency (LaMf), with four peaks at 36 °C per week; (6) Low amplitude–Low frequency (LaLf), with one peak at 36 °C per week. Each treatment had 10 replications. All the experiments were conducted in climate-controlled growth chambers (75 ± 5% RH, 14 h:10 h L:D photoperiod).

### 2.3. Host Plant Infestation and Whitefly Nymph Preparation

To generate a batch of uniform whitefly nymphs, four clean cotton plants with four fully open leaves were transferred into a cage (25 × 25 × 25 cm) and covered with fine gauze. Then, 200 adult whiteflies (less than 2 days old) were transferred to this cage. The whiteflies were able to lay eggs for 24 h before being removed. The cotton leaves and eggs were incubated at 26 °C for 10 days and then checked daily with a binocular microscope until the whiteflies reached the appropriate developmental stage. According to previous experiments, the late 2nd- to early 3rd-instar stage of the host nymph was the appropriate stage [14]. We found that this stage occurred after 15–17 days. Finally, these cotton plants were used for the next experiment. We set up six treatments; each treatment had 10 replicates, so at least 240 cotton plants were prepared for these experiments.

### 2.4. Simulated Heat Wave Experiment

Four cotton plants infested with suitable nymphs were placed into a cage (25 × 25 × 25 cm). Five newly emerged and mated *E. hayati* females were then released into the cage, and this cage was moved into a climate-controlled growth chamber set to the temperature regimes above. After 7 days, all the cages with cotton plants were moved into a growth chamber at a constant temperature of 26 °C (75 ± 5% RH, 14 h:10 h L:D photoperiod) for continuous observation. Fecundity and host-feeding event by the parasitoids in each cage were recorded after 8 days. If host mycetome displacement was visible though the cuticle under a microscope 8 days after the egg was laid, the host was considered parasitized by parasitoid and positive fecundity was recorded. If the host body appeared flat and desiccated, it meant that the host was consumed directly and host-feeding event was recorded [14]. Furthermore, we recorded the number of emerged parasitoids and adult whiteflies, and sexed the parasitoids. Finally, the cotton leaves were detached from the cotton plant, and the leaf area of each leaf was measured by a portable area meter (LI-3000C, LI-COR Company, Lincoln, Nebraska, USA). The total leaf area per cage was calculated as the sum of each leaf area in a cage. We obtained the total host number per cage by adding the emerged whitefly number to the total host mortality. Total host mortality was equal to lifetime fecundity added host-feeding event, as natural host mortality was rare. Then, host density was calculated by the total host number per cage divided by the total leaf area, and biocontrol efficiency was calculated by the total host mortality divided by the total host number.

### 2.5. Statistical Analysis

A general linear model (GLM, two-way ANCOVA) was used to test for differences in life history traits (fecundity, host feeding, total whitefly mortality, biocontrol efficiency, female offspring, male offspring, total emerged offspring, and sex ratio of offspring) with heat waves with different frequencies and amplitudes. Heat wave frequency (seven times/week, four times/week and one time/week) and amplitude (36 °C and 42 °C) were considered as two factors, and host density was considered as a covariate. As all life history traits met the assumptions of normality and homoscedasticity, Tukey’s multiple comparisons test between least square means was conducted when any of the factors in the models were significant.

If the host density was not significant as a covariate, we excluded it and changed the general linear model to two-way ANOVA. Heat wave frequency and amplitude were considered as two factors. Tukey’s multiple comparisons test between least square means was conducted when any of the factors in the models were significant.

All the raw data of life history traits per replication were averaged and divided by five before analysis as we released five newly emerged female parasitoids into a cage as one replication in this study. Both the biocontrol efficiency and sex ratio of offspring were percentage data, so these data were ARCSIN transformed before analysis.

All of the analyses were conducted using SAS software (version 9.20).

## 3. Results

### 3.1. Life History Traits and Biocontrol Efficiency

#### 3.1.1. Host-Feeding Event

The host-feeding event of parasitoids was significantly affected by host density (*F*_1,59_ = 6.68, *p* = 0.0127, Figure 2a), heat wave frequency (*F*_2,59_ = 6.55, *p* = 0.003, Figure 2a) and heat amplitude (*F*_1,59_ = 7.93, *p* = 0.0069, Figure 2a). There was no interaction between heat wave frequency and heat amplitude (*F*_2,59_ = 2.53, *p* = 0.0894, Figure 2a). When the heat amplitude was high (42 °C), the host-feeding event of parasitoids facing low heat wave frequency was significantly larger than those facing high heat wave frequency (Tukey-Kramer test: *p* = 0.0345). When the heat amplitude was low (36 °C), the host-feeding event of parasitoids facing different heat wave frequencies showed no difference (Tukey-Kramer test: *p* > 0.05 for all). When the heat wave frequency was fixed, a significant difference in the host-feeding event was only generated at a high heat wave frequency (Tukey-Kramer test: *p* < 0.05).

#### 3.1.2. Fecundity

Only heat amplitude significantly affected the fecundity of parasitoids (*F*_1,59_ = 11.02, *p* = 0.0016, Figure 2b); there was a significant interaction between heat wave frequency and heat amplitude (*F*_2,59_ = 2.38, *p* = 0.0424, Figure 2b). When the heat amplitude was fixed at a high level (42 °C), the heat wave frequency substantially changed the lifetime fecundity of parasitoids (Tukey-Kramer test: *p* < 0.05 for all, except for between high frequency and medium frequency). When the heat wave frequency was fixed, the fecundity of parasitoids under high heat frequency (Tukey-Kramer test: *p* < 0.05) and medium heat frequency (Tukey-Kramer test: *p* < 0.05) conditions showed significant differences.

#### 3.1.3. Adult Longevity

Both heat wave frequency (*F*_2,59_ = 36.92, *p* < 0.0001, Figure 2c) and heat amplitude (*F*_1,59_ = 67.89, *p* < 0.0001, Figure 2c) significantly affected the longevity of adult female parasitoids, and there was an interaction between them (*F*_2,59_ = 3.94, *p* = 0.0252, Figure 2c). When the heat amplitude was fixed at a high level (42 °C), the heat wave frequency substantially changed the adult female longevity of parasitoids (Tukey-Kramer test: *p* < 0.05 for all, except for between high frequency and medium frequency). When the heat amplitude was fixed at a low level (36 °C), a significant difference in the adult longevity of parasitoids was found only between low frequency and high frequency (Tukey-Kramer test: *p* = 0.0009). When the heat wave frequency was fixed, a difference in female adult longevity between two heat amplitude groups was generated at high frequency (Tukey-Kramer test: *p* < 0.0001) and medium frequency (Tukey-Kramer test: *p* < 0.0001).

#### 3.1.4. Total Host Mortality

Similar to fecundity, only heat amplitude significantly affected the total host mortality of parasitoids (*F*_1,59_ = 10.79, *p* = 0.0018, Figure 2d), and there was a significant interaction between heat wave frequency and heat amplitude (*F*_2,59_ = 2.40, *p* = 0.015, Figure 2d). When the heat amplitude was fixed at a high level (42 °C), the heat wave frequency largely changed the total host mortality of parasitoids (Tukey-Kramer test: *p* < 0.05 for all, except for between high frequency and medium frequency). When the heat wave frequency was fixed, the total host mortality of parasitoids under high heat frequency (Tukey-Kramer test: *p* < 0.05) and medium heat frequency (Tukey-Kramer test: *p* < 0.05) conditions showed significant differences.

#### 3.1.5. Biocontrol Efficiency

Host density (*F*_1,59_ = 10.06, *p* = 0.0026, Figure 2e) and heat amplitude (*F*_1,59_ = 4.11, *p* = 0.048, Figure 2e) significantly affected the biocontrol efficiency of the parasitoids, but heat wave frequency (*F*_2,59_ = 2.71, *p* = 0.0761, Figure 2e) did not, and there was an interaction between heat wave frequency and heat amplitude (*F*_2,59_ = 4.26, *p* = 0.0195, Figure 2e). When the heat amplitude was high, the biocontrol efficiency of parasitoids under low heat wave frequency conditions was significantly greater than those under high and/or medium heat wave frequency conditions (Tukey-Kramer test: *p* < 0.05 for both). When the heat amplitude was low, the biocontrol efficiency of parasitoids under different heat wave frequencies showed no difference (Tukey-Kramer test: *p* > 0.05 for all). When the heat wave frequency was fixed, a significant difference in biocontrol efficiency was generated at only a high heat wave frequency (Tukey-Kramer test: *p* = 0.0431).

### 3.2. Emerged Offspring Number (Female, Male) and Sex Ratio

#### 3.2.1. Female Offspring

Heat wave frequency significantly affected the number of female offspring (*F*_2,59_ =3.94, *p* = 0.0254, Figure 3a), while heat wave amplitude did not (*F*_1,59_ = 3.51, *p* = 0.0664, Figure 3a), and there was no interaction between heat wave frequency and amplitude (*F*_2,59_ = 0.29, *p* = 0.7596, Figure 3a). When the heat amplitude was fixed, the emerged female parasitoids under low-heat wave frequency condition were significantly larger than those under high-heat wave frequency conditions, but emerged females in both the low and high-heat wave frequencies showed no significant differences from those under medium-heat wave frequency conditions, regardless of high or low heat amplitude (Tukey-Kramer test: *p* < 0.05 for both).

#### 3.2.2. Male Offspring

Heat amplitude significantly affected the number of male offspring (*F*_1,59_ =17.42, *p* < 0.0001, Figure 3b), but the heat wave frequency did not (*F*_2,59_ = 0.81, *p* = 0.8355, Figure 3b); there was an interaction between heat wave frequency and amplitude (*F*_2,59_ = 9.98, *p* = 0.0002, Figure 3b). When the heat amplitude was high, the number of male offspring in the low heat wave frequency group was significantly higher than that in the high heat wave frequency group (Tukey-Kramer test: *p* =0.0168), but was not different from that in the medium heat wave frequency group (Tukey-Kramer test: *p* > 0.05). When the heat amplitude was low, the number of male offspring in the high heat wave frequency group was significantly different from that in the low heat frequency group (Tukey-Kramer test: *p* = 0.045). When the heat wave frequency was fixed, a significant difference between two amplitudes was only generated at high and medium heat wave frequencies (high frequency: *p* < 0.0001, medium frequency: *p* = 0.0084).

#### 3.2.3. Sex Ratio of Offspring

Both heat wave frequency (*F*_2,59_ = 3.21, *p* = 0.0484, Figure 3c) and heat amplitude (*F*_1,59_ = 9.33, *p* = 0.0035, Figure 3c) affected the sex ratio of offspring, and there was a significant interaction (*F*_2,59_ = 14.23, *p* < 0.0001, Figure 3c). When the heat amplitude was low, the sex ratio of parasitoid offspring in the low heat wave frequency group (LaLf, female:male = 54.6:24.4) was significantly higher than those in the high (LaHf, female:male = 41.6:39.2) or medium (LaMf, female:male = 51.7:37.9) heat wave frequency groups (high frequency: *p* < 0.0001, medium frequency: *p* = 0.0283), while there was no difference when the heat amplitude was high (Tukey-Kramer test: *p* > 0.05 for both). When the heat frequency was fixed, significant differences between high heat amplitude and low heat amplitude were generated in only the high frequency (HaHf, female:male = 25.9:13; Tukey-Kramer test: *p* =0.0010) and medium frequency groups (HaMf, female:male = 42.5:18.9; Tukey-Kramer test: *p* = 0.0098).

## 4. Discussion

The frequency and amplitude of extreme climatic events are predicted to increase with global warming and have greater impacts on ecosystems than increasing mean temperatures [17]. However, the effects of extreme heat wave events on organisms, especially on secondary trophic organisms, such as parasitic insects, have rarely been studied [3,9,10]. In the present study, we chose a simple plant (cotton)–herbivore (*B. tabaci*)–parasitoid (*E. hayati*) system to explore the consequences on life history traits and biocontrol efficiency of a host-feeding parasitoid through simulated heat waves as a function of amplitude and frequency. Our results showed that heat waves significantly affected the life history traits and biocontrol efficiency of *E. hayati*. Moreover, the biocontrol efficiency also covaried with host density.

Heat amplitude significantly affected the life history traits of *E. hayati*. At low amplitude (36 °C), most life history traits (except for longevity) did not show significant differences. This means that if the heat amplitude is not sufficiently high, the heat frequency will not affect the life history traits of the parasitoid. In fact, *E. hayati* may exhibit heat stress at a low heat wave amplitude (36 °C), as Zhang et al. [15] confirmed that *E. hayati* females encountered heat stress when the environmental temperature was constantly maintained at 34 °C, and the parasitoid fitness largely decreased. However, the heat stress in the present study was inconsistent; *E. hayati* coped with heat stress because they had sufficient time to recover or move to a more comfortable microclimate to escape heat conditions [9,15].

When the heat amplitude was high (42 °C), all of the life history traits of *E. hayati*, including longevity, showed significant differences, which decreased with increasing heat frequency. When the frequency of the heat wave was low (1 time/week), even when the amplitude of the heat wave was high (42 °C), the life history traits of *E. hayati* were not significantly affected. However, when the frequency of heat waves was increased to medium (4 times/week) or high (7 times/week), those parameters were negatively affected. Schreven et al. [8] indicated a similar result in the endoparasitoid *D. semiclausum*. A low heat pulse in which the heat amplitude was not over the optimal temperature had no detrimental effect on parasitoid fitness, but a high heat pulse (over the optimal temperature) negatively affected these parameters. Unfortunately, they did not take the frequency of heat waves into consideration. Gillespie et al. [18] found that simulated heat waves had quite different effects on the fitness of two aphid parasitoid species in the same guild. *Aphidius abdominalis* Dalm. (Hymenoptera: Aphidiidae) was not impaired by high-frequency and high-amplitude treatments. In contrast, *Aphidius matricariae* Haliday (Hymenoptera: Braconidae) experienced delayed development, reduced fecundity and/or increased mortality under the high-frequency and high-amplitude treatments. The foraging ability and fecundity of *A. matricariae* were not affected by simulated heat waves as a function of heat amplitude (32 °C vs. 40 °C) and frequency (two times/week vs. seven times/week) [9]. These contrasting results could be primarily explained by species-specific life cycles, host availability, and interactions of functional responses and reproductive capacity in the different parasitoid species [2]. However, Roux et al. [19] showed that the survival and reproduction rates of the aphid parasitoid *Aphidius avenae* Haliday (Hymenoptera: Aphidiidae) was negatively affected by a 1 h heat shock. This could be because these parasitoids were directly exposed to heat stress in glass tubes without any shelter. All the other parasitoids in the studies mentioned above were released into a fixed cage or box containing previously placed host plants and host herbivorous insects. Hence, when extreme heat events occurred, the adult parasitoids could fly or walk to escape the heat by moving to a more comfortable microenvironment to mitigate the detrimental influence of extreme heat events [8,9,18].

In addition, high heat amplitude and frequency may result in the consumption of more nutritional substances, as the rising environmental temperature can affect body temperature and accelerate the metabolic rate [20,21]. In other words, an increased amount of energic substances may be consumed, and a decreased amount of capital resources, which means total nutrient substances in a newly emerged parasitoid may be allocated to oogenesis and body maintenance [22]. The main nutritional substances in an emerging parasitoid, such as lipids, are constant and incapable of synthesis during the adult stage [23,24]. *E. hayati* under high temperature (42 °C) conditions can experience effects in a shorter time and require more time to recover than those under low temperature (36 °C) conditions [25]. Aparasitoid exposed to a high temperature (42 °C) usually requires 20–30 min to revive, while a parasitoid exposed to a relatively low temperature (36 °C) can recover in 10 min [25]. Taking this into consideration, when parasitoids experience a high temperature event, they not only need increased nutritional substances to maintain their life, but also need increased time to cope with high-stress situations. That is why the life history traits of *E. hayati* (lifetime fecundity, total host mortality, biocontrol efficiency and longevity) in the present study significantly decreased with heat amplitude and heat frequency.

Heat waves can not only detrimentally affect the life history traits of adult parasitoids but also negatively affect the emerged number of offspring. When the heat amplitude was 42 °C, the number of female and male parasitoids significantly decreased with increasing heat frequency. Moreover, we found that heat frequency changed the sex ratio of offspring when the heat amplitude was low, but did not change it when the heat amplitude was high. Few studies have investigated the effect of heat waves on the sex ratio of parasitoids [2]. In previous studies, researchers mainly focused on the effect of extreme low temperatures or events on the sex ratio of parasitoids [26,27], as low temperatures sterilize males, decrease the rate of movement of either sex to a point at which mating no longer occurs, and incapacitates sperm [26,28]. Additionally, some previous studies confirmed that high temperatures can negatively influence the life traits of *B. tabaci* [29], as temperatures change host metabolism and immune system activity by modulating the genes encoding ferritin and HSP [30,31]. Hence, we expected that the negative influences of the life history traits of *E. hayati*, especially the emerged number of their offspring, would partly result from the effect of heat waves on the ecophysiological response of the host. However, this result was uncertain. The way in which heat waves affect the sex ratio of parasitoid offspring should be further investigated in the future.

We used two-way ANCOVA to analyze all of the life history traits of parasitoids. However, only host feeding and biocontrol efficiency showed significant relationships with host density. Thus, different host densities can significantly affect the host-feeding events and biocontrol efficiency of *E. hayati*. Generally, host availability and host density affect the parasitoid’s foraging and host selection behavior [12]. Zhang et al. [14] stated that *E. hayati* females preferred to feed on early instar nymphs of *B. tabaci* but could parasitize all nymph stages of *B. tabaci*. That is, parasitoids have a shorter time window for feeding on hosts than for parasitizing hosts. Hence, it would be more difficult to search for a suitable host for feeding than for parasitizing under extreme heat wave conditions. This could be the main reason why host density affected the number of host-feeding events in *E. hayati*, further affecting the biocontrol efficiency.

## 5. Conclusions

Parasitoids play a critical role as biological control agents in agriculture worldwide. Hence, understanding how heat wave frequency and amplitude changes are linked to the life history traits and biocontrol efficiency of a host-feeding parasitoid has significant implications for utilizing these biocontrol agents under a changing climate. Our results found that both the amplitude and frequency of heat waves can significantly affect adult phenotypes. Under low heat wave amplitude, the frequency of the heat wave did not change the life history trait of the parasitoid; however, when the heat amplitude was 42 °C, medium (four times/week) and high frequencies (seven times/week) of heat waves detrimentally affected these parameters. Hence, these findings suggest that to obtain optimal biological control efficiency in a parasitoid, we need to carefully monitor heat wave weather (especially the amplitude and frequency) over the short term (usually 7–10 days) before releasing a host-feeding parasitoid. If a heat wave event is predicted, it may be better to change or delay the time of release of a parasitic natural enemy to avoid these adverse conditions.

## Figures and Tables

**Figure 1 insects-10-00419-f001:**
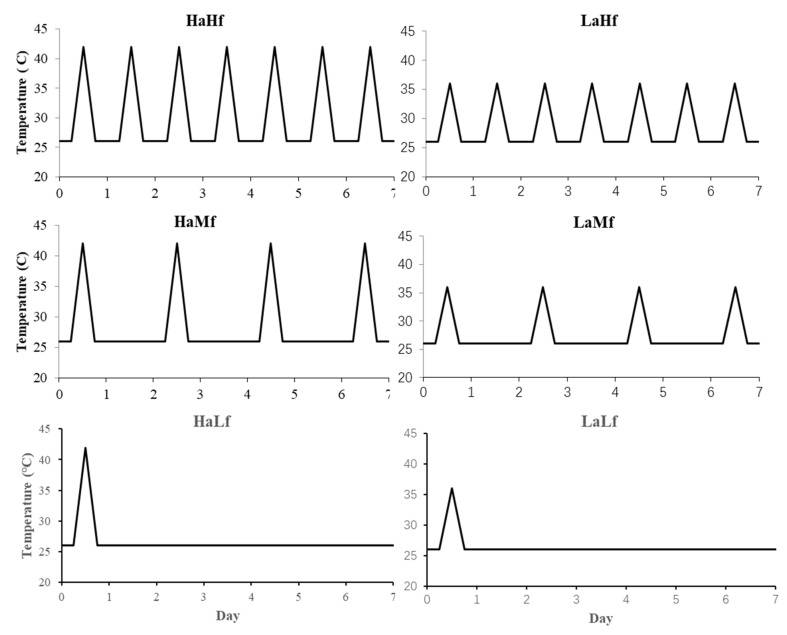
The temperature regimes generated by a combination of the amplitude and frequency of temperature peaks. ‘HaHf’ means high amplitude and high frequency, ‘HaMf’ means high amplitude and medium frequency, ‘HaLf’ means high amplitude and low frequency, ‘LaHf’ means low amplitude and high frequency, ‘LaMf’ means low amplitude and medium frequency, and ‘LaLf’ means low amplitude and low frequency.

**Figure 2 insects-10-00419-f002:**
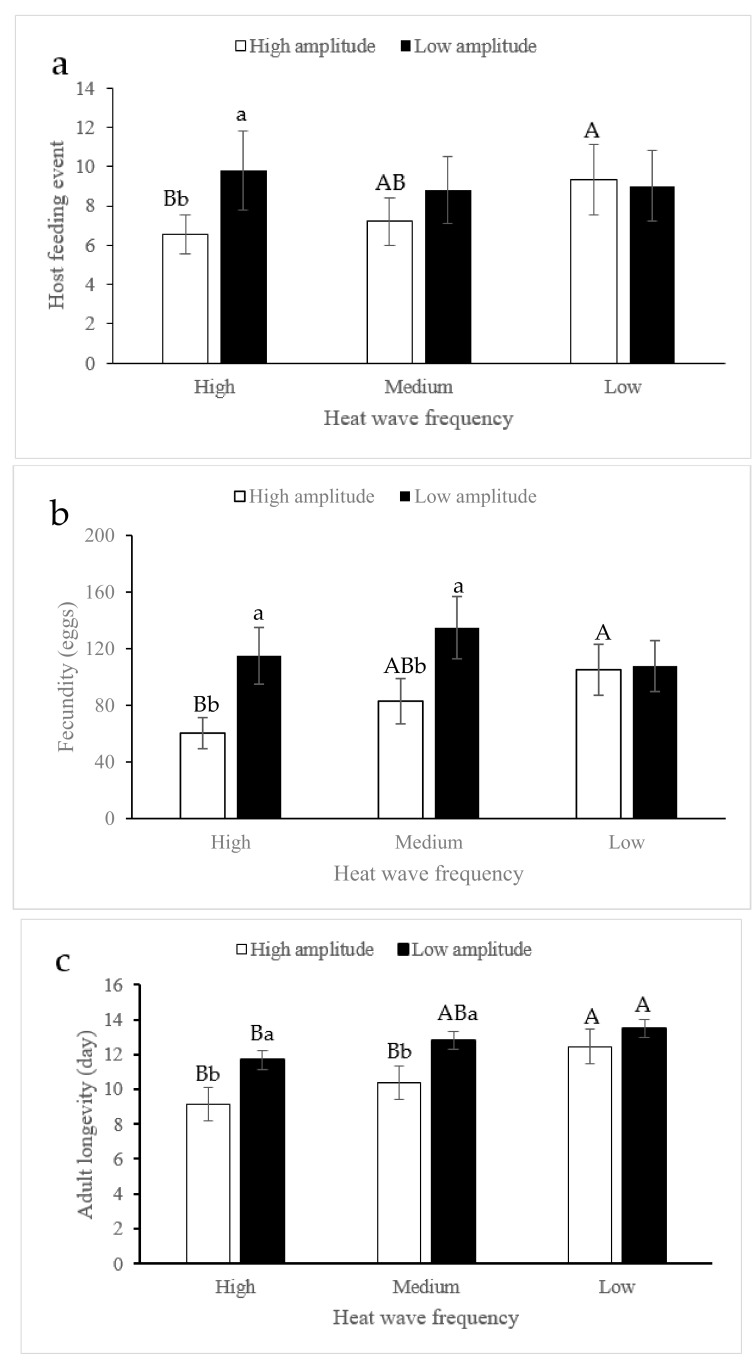
The effect of simulated heat waves on life history traits of *Eretmocerus hayati* ((**a**) host feeding, (**b**) fecundity, (**c**) adult longevity, (**d**) total host mortality, and (**e**) biocontrol efficiency). All traits are presented as means with SEMs. Bars topped by different capital letters within the same amplitude of heat waves indicate a significant difference between different frequencies; the different lowercase letters indicate that there was a significant difference in the same frequency between different amplitudes of heat waves (two-way ANOVA).

**Figure 3 insects-10-00419-f003:**
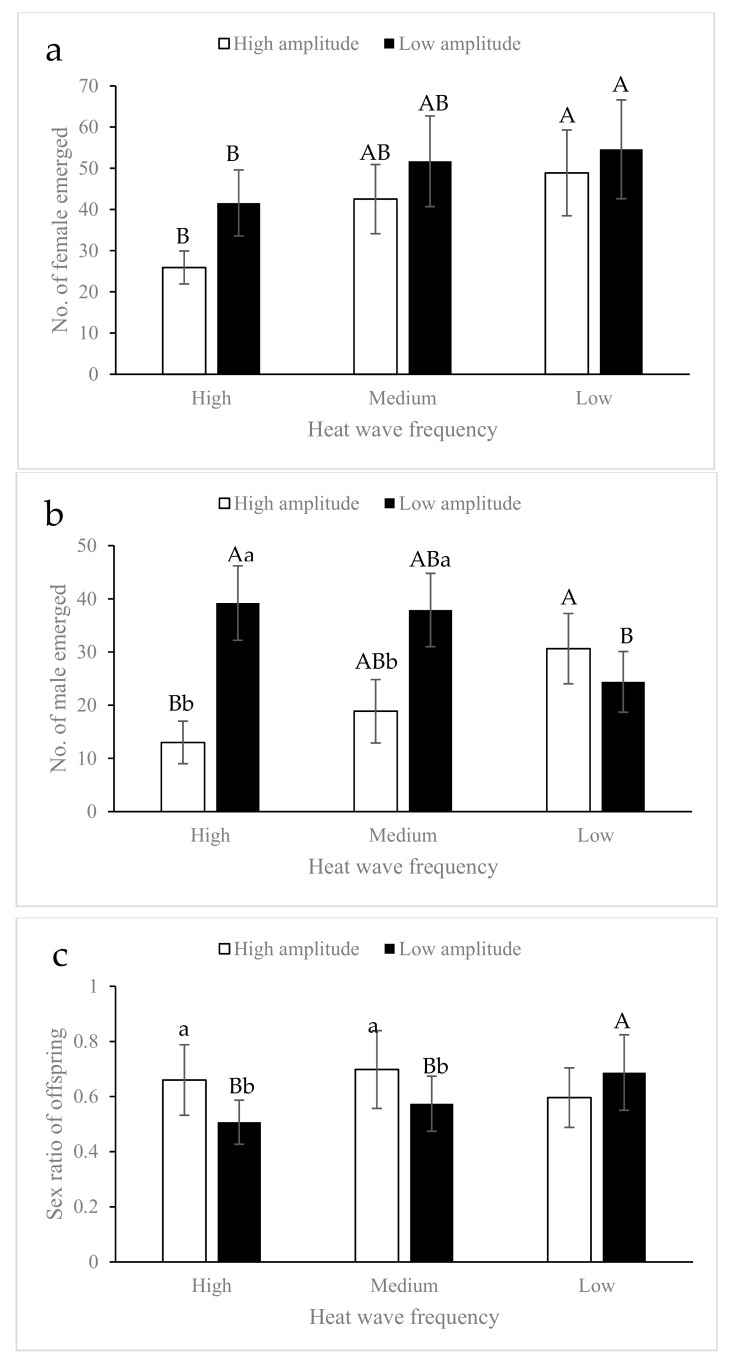
The effect of simulated heat waves on the offspring fitness of *Eretmocerus hayati* (**a**) No. of females emerged, (**b**) No. of males emerged, (**c**) sex ratio of offspring). All the fitness traits are presented as means with SEMs. Bars topped by different capital letters within the same heat wave amplitude indicate significant differences between different frequencies; the different lowercase letters indicate that there was a significant difference in the same frequency between different amplitudes of heat waves (two-way ANOVA).

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
