# Peer review of "Effects of Simulated Heat Waves on Life History Traits of a Host Feeding Parasitoid"

_insects, 2019, doi:10.3390/insects10120419_

Round 1
Reviewer 1 Report
The manuscript entitled „Effects of simulated heat waves on life history traits of a host feeding parasitoid” present important studies about role of heat waves, related to global warming, in shaping of ecophysiology of parasitoid Eretmocerus hayati. The reviewed manuscript is well written and well-designed, but some minor remarks should be corrected before acceptance of this manuscript to publication.
General questions
I think that some of the obtained results probably are the effect of ecophysiological response of host on heat waves. Research concern this issue showed that high temperature negatively influences on life traits of B. tabaci (Curnutte i wsp., 2014; Diaz i wsp., 2015; Mahadav i wsp., 2009). Of course, Authors present results of influence of heat waves on host mortality. However, probably presented in the manuscript differences in E. hayati life trait may be strictly related to changes in host metabolism and immune system activity (by modulation of genes encoding ferritin and HSP) (Diaz i wsp., 2015; Mahadav i wsp., 2009). In my opinion, Authors should mention about these probable dependencies in the Introduction and Discussion section.
I wonder, whether Authors also analysed fecundity of F1? Maybe the heat waves have significant influence of F1 reproduction. I also wonder, how tested species will be reacted on mixed amplitude, which may be also observe in natural environmental. Of course, these two questions are not a remarks and are the results of my personal interest of this topic.
Minor remarks
Lines 75-76. “To generate a laboratory colony, E. hayati was collected from cotton fields during the summer of 2012 (…)”. Its means that the same population of individuals are breeding incessantly from 2012. Whether from this time, some new individuals captured in natural environmental, were introduced to culture through avoiding inbreeding?
Figures. Figure captions should contain more information about presented data, especially whether data are presented as a mean/median with SD/SEM.
3.2.1 Female offspring section. The results presented in this section should be better described. Only comparison this section with mentioned Figure 3a allow to appropriate interpretation of results.
Line 212. “Heat amplitude significantly affected the number of female offspring”. Should be male offspring.
Section 3.2.3. Sex ratio of offspring and Figure 3c. To better understand results, Authors should add also specific information about percentage ratio of male and females. Presenting only value of sex ratio is not fully informative.
Lines 261-262. “It was a pity that they did not take the frequency of heat wave into consideration.” In my opinion this sentence should be a little bit mitigated.
References
Curnutte, L. B., Simmons, A. M. i Abd-Rabou, S. (2014). Climate change and Bemisia tabaci (Hemiptera: Aleyrodidae): impacts of temperature and carbon dioxide on life history. Annals of the Entomological Society of America 107, 933-943.
Diaz, F., Orobio, R. F., Chavarriaga, P. i Toro-Perea, N. (2015). Differential expression patterns among heat-shock protein genes and thermal responses in the whitefly Bemisia tabaci (MEAM 1). Journal of Thermal Biology 52, 199-207.
Mahadav, A., Kontsedalov, S., Czosnek, H. i Ghanim, M. (2009). Thermotolerance and gene expression following heat stress in the whitefly Bemisia tabaci B and Q biotypes. Insect Biochemistry and Molecular Biology 39, 668-676
Author Response
Dear reviewer,
Many thanks for your attentions.
We have revised entire manuscript accroding to your comments. A point-by-point response to the comments is sending to you by the attachment, please check it.
All the best,
Yibo Zhang

Reviewer 2 Report
REVIEW on the Effects of simulated heat waves on life history traits 2 of a host feeding parasitoid by
Yi-Bo Zhang, An-Pei Yang, Gui-Fen Zhang, Wan-Xue Liu and Fang-Hao Wan Comments to the Authors:
Yi-Bo Zhang et al. report their results on the biological control consequences of a host feeding parasitoid Eretmocerus hayati. Investigations of relationships among responsiveness to extreme temperature events and the life history traits are actually important in current climatic researches, both in bioclimatological and agroecological studies.
This is an interesting manuscript, which should become a relevant contribution to Insects after a thorough revision.
First, the ms needs a thorough linguistic revision, as in includes a large number of typos and grammatically incorrect phrases. To facilitate such a revision, I include examples for linguistic inconsistencies:
Introduction:
Page 1 Line 33: insert an ‘s’ after wave in the bracket.
Page 2 Line 44: ‘files’ should be replaced by ‘flies’ in the brackets.
Materials and methods:
Page 2 Line 68: what do you mean by RH and L:D?
Page 2 Line 70: what does MED mean?
Please elaborate on the measurement procedure of life history traits in the materials and methods section to clarify the results.
Results:
Page 4. Line 123: ‘8 d’ means 8 day? Please specify.
Page 4. Line 154: ’Heat feeding of…’ should be ’Host feeding of…’
What do you mean by ‘host feeding event’ on Figure 2? What type of measured data did you use? This is not clear for me in the ms, please include it in the text.
Page 5 Line 167: How do you measure fecundity? Please elaborate this in the materials and methods section.
Page 5 Line 183: What kind of measured data did you use here?
Page 5 Line 192: Host density and heat amplitude significantly affected the biocontrol efficiency of parasitoids, but heat wave frequency did not, there was interaction between heat wave frequency and heat amplitude.
Page 5 Line 192: What do you mean by biocontrol efficiency? Please clarify in the ms. What kind of measured data did you use here
Pagae 7 Line 207: replace ‘not’ with ‘no’.
….there was no interaction between what?
Pagae 7 Line 209: ‘When heat amplitude was fixed, the emerged female 208 parasitoids in the low heat wave frequency were significant larger than those in the high heat wave 209 frequency regardless of high or low heat amplitude’
Page 7 Line 212: replace ‘female’ by male.
Page 7 Line 213: ….there was interaction between what?
Page 7 Line 218: …..was significantly difference in what?
Paragraph 3.2: In this part of the investigation what kind of measurements did you use? Please clarify it. You write that “When heat amplitude was high, the male offspring of parasitoids in low heat wave frequency was significantly larger than those in high heat wave frequency” The offspring were larger or the number of the offspring? (also see figure 3, when you write ‘No. male emerged’)
Page 8 Line 224: ….. was significantly interaction between what?
Page 8 Line 229: Unclear sentence: ‘When heat frequency was fixed, significant differences between high amplitude and low amplitude were only generated at high frequency and medium frequency.
Discussion and conclusion:
Page 9 Line 242: You write that ‘Our results showed that heat wave significantly affected the life history traits and biocontrol efficiency of E. hayati’, but you do not mention anything about the examination of the biocontrol efficiency in the result section.
Page 9 Line 247: insert that before ‘could’
Page 10 Line 295: ‘with heat frequency increased’ change by ‘with increasing heat frequency’
Figures:
Figure 2 what do the different letters mean (a, b, A, B)? What do you mean by ‘host feeding event’?
Author Response
Dear reviewer,
Many thanks for your attentions.
We have revised the manuscript according to your comments. A point-by-point response to your comments was sending to you by the attachment, please take time to check it.
All the best,
Yibo Zhang

Reviewer 3 Report
Please see my edits/comments in the attached file.

Author Response
Dear reviewer,
Many thanks for your attentions.
Because most of your comments were about language problems, we accepted all of your suggests and revised our manuscript according to your comments. So, we did not write a point to point response to your comments, hope you can understand. Meanwhile, we invited a native English entomologist to check our revised manuscript for solving the language problems.
All the best,
Yibo Zhang
Round 2
Reviewer 2 Report
Comments to the Authors:
The ms has significantly improved after the revision. However, I found a small number of issues to be corrected prior to acceptance, as listed below.
You mentioned that ‘Host feeding event means the number of hosts killed by parasitoids as directly feeding way.’ Please incorporate it in the text or in the figure legend.
In Line 137-139 of Page 4.: there is no description in this paragraph how did you measure fecundity.
In Line 146-147, Page 4. You wrote that ‘Total host mortality was used here. We described the measure way of this’, but I cannot find it in the text here.
You wrote that ‘We described the measure way of biocontrol efficiency in Line 148-149, Page 4.’ In this part of the ms there is no information on it. And it is still not understandable for me, what do you mean by biocontrol efficiency. Please clarify it.
Author Response
Q1:You mentioned that ‘Host feeding event means the number of hosts killed by parasitoids as directly feeding way.’ Please incorporate it in the text or in the figure legend.
Response:we incorporated this sentence in the text (Line129).
Q2:In Line 137-139 of Page 4.: there is no description in this paragraph how did you measure fecundity.
Response:Sorry, we described how to measure fecundity in Line 125-127, please check it. Actually, we did describe this at Line 137-139 of page4 in revised manuscript with track version. You may check the revised manuscript without track version, so the information was mismatched.
Q3:In Line 146-147, Page 4. You wrote that ‘Total host mortality was used here. We described the measure way of this’, but I cannot find it in the text here.
Response:Similar to Q2, we described how to measure total host mortality in Line 133-135. I think the reason could be the same to Q2. Actually, in this study, total host mortality was equal to lifetime fecundity added host-feeding events, as natural host mortality was rare.
Q5:You wrote that ‘We described the measure way of biocontrol efficiency in Line 148-149, Page 4.’ In this part of the ms there is no information on it. And it is still not understandable for me, what do you mean by biocontrol efficiency. Please clarify it.
Response:Sorry, we described how to measure biocontrol efficiency in Line 136-137. I think the reason could also be the same to Q2 and Q3. In the study, we defined biocontrol efficiency as the total host mortality divided by the total host number. Actually, it was just a percentage of the total death host number, containing the host fed and parasitized by parasitoid, in total host number.